# Improved Differentiation Towards Insulin Producing Beta-Cells Derived from Healthy Canine Pancreatic Ductal Organoids

**DOI:** 10.3390/vetsci12040362

**Published:** 2025-04-13

**Authors:** Boyd H. T. Gouw, Flavia C. M. Oliveira, Hans S. Kooistra, Bart Spee, Lisa van Uden, Louis C. Penning

**Affiliations:** 1Department of Clinical Sciences, Faculty of Veterinary Medicine, Utrecht University, 3584 CM Utrecht, The Netherlands; b.h.t.gouw@students.uu.nl (B.H.T.G.); flavia.matos@ufra.edu.br (F.C.M.O.); h.s.kooistra@uu.nl (H.S.K.);; 2Health and Animal Production in Amazônia Program, Universidade Federal Rural da Amazônia, Belém 66077-830, Brazil

**Keywords:** diabetes mellitus, pancreatic differentiation, pancreatic stem cells, dog, organoids

## Abstract

Insulin is a crucial hormone that regulates blood glucose levels. Both too-high and too-low blood glucose levels can be deleterious. Diabetes mellitus (DM) is a disease where the insulin levels, and thus blood glucose levels, are not properly maintained. This disease occurs in humans and pets roughly at the same percentage. One of the causes of DM is related to the destruction of the cells that produce insulin upon high blood glucose levels, viz. the beta-cells in the pancreas. Current treatment of DM in dogs is based on twice-daily injections of insulin. This greatly affects the quality of life of both the dog and its owner. Therefore, alternatives for this treatment are needed and are under investigation. One of the strategies is to replace damaged beta-cells with novel beta-cells. This manuscript describes a way to culture beta-cells derived from pancreas stem cells obtained from healthy dog pancreas. We used cadaveric material, not privately owned animals, in line with the University 3R policy to create functional beta-cells. Although far from application in cell replacement therapy, these cells provide novel means to investigate DM in a cell culture system, without the need to work on living dogs.

## 1. Introduction

Diabetes mellitus (DM) is a common endocrine disorder in humans and other mammals, but it also affects some birds and reptile species. DM is caused by various pathologies, all of which lead to persistent hyperglycemia, a deleterious condition potentially resulting in diabetic neuropathy, diabetic cataracts, and, if not monitored and treated, life-threatening complications.

Recent studies have shown that 1 in every 300 dogs, and even more for cats, is diagnosed with DM; the prevalences are lower compared to those in humans, but underdiagnosis in pets is likely [1,2].

Age, breed, gender, and lifestyle are factors contributing to the development of DM. The common age of diagnosis of DM in dogs is 7–9 years old, and in cats, it is 10–13 years old.

Diabetes is more prevalent in female dogs than in male dogs and almost twice as likely in male cats than in female cats [3,4]. Also, Burmese cats are more likely to develop DM, while dog breeds such as the Alaskan Malamute, Keeshond, or Samoyed are more prone to be diagnosed with DM [4]. Dogs suffer mostly from type 1 diabetes (DT1, in humans, often regarded as insulin dependent, early onset, autoimmune destruction of β-cells), whereas cats suffer mostly from type 2 diabetes (DT2, late-onset, insulin insensitivity); finally, type 3 (DT3, insulin resistance due to other disorders or drugs) and type 4 (DT4, gestational induced) are significantly less common in pets [1,2,4].

Despite over a century of research on dogs and DM, the current treatment options for dogs are often deemed inadequate by their owners, as exemplified by the large number of owners opting for euthanasia after a diagnosis of DM [5]. This prompted many researchers in the veterinary medicine field to investigate a more sustainable and less labor-intensive treatment, although this is a long-shot and insulin will likely be the main therapy.

In the field of Regenerative Medicine (RM) and stem cell biology, both induced Pluripotent Stem Cells (iPSCs) and Adult Stem Cells (ASCs) have become the preferred cell types for cell-based therapies. Although the use of large animals has significant pre-clinical implications in RM, disease modelling and drug screening are practically challenging [6,7]. Here, 3D organoid cultures, defined as containing several cell types that develop from stem cells or organ progenitors and self-organize through cell sorting and spatially restricted lineage commitment similar to the process in vivo, can be instrumental [8]. As 3D organoid cultures from ASCs from canine intestine, liver, and other organs are established and characterized (readers are referred to reviews described in detail), establishing canine pancreatic organoids from pancreatic ductal stem cells (source for the ACSs) seemed a logical approach [9,10,11,12,13,14]. Therefore, the aim of this study was to establish organoid cultures consisting of differentiated β-cells from canine pancreatic ductal cells. To achieve this, we have adjusted the existing protocol for human β-cell differentiation to freshly isolated canine ductular pancreatic stem cells. Here, we show that a step-by-step differentiation protocol, mimicking the pancreatic organogenesis, resulted in 3D cultures of glucose-responsive immature β-cells. It is unknown if stem cells from diseased animals will behave similarly.

## 2. Materials and Methods

### 2.1. Cell Culture

#### 2.1.1. Pancreatic Tissue Samples

Pancreatic tissue was obtained from surplus material of healthy dogs (n = 4), euthanized for non-pancreatic-related research, in accordance with ethical standards in accordance with the Utrecht University’s 3R policy. From one dog, the owners allowed us to use pancreatic material after informed consent. For information about the donors, see Appendix A. The collected pancreatic tissue samples were placed in 50 mL tubes containing ice-cold Advanced DMEM/F12 (Dulbecco’s Modified Eagle Medium, Thermo Fisher Scientific, Waltham, MA, USA), hereafter referred to as A/D, supplemented with 1% (*v*/*v*) penicillin–streptomycin (pen-strep, Gibco, Thermo Fisher Scientific), 1% (*v*/*v*) HEPES (Gibco, Thermo Fisher Scientific), and 1% (*v*/*v*) GlutaMax (Gibco, Thermo Fisher Scientific), hereafter referred to as A/D+. The tubes were transported to the laboratory, while keeping the tubes continuously on ice prior to processing. The obtained samples were frozen in Recovery™ Cell Culture Freezing Medium (Gibco, Thermo Fisher Scientific) for future usage or used directly for organoid isolation (Figure 1A).

#### 2.1.2. Organoid Isolation

Fresh digestion medium was made by adding collagenase II (0.3 mg/mL; Gibco, Thermo Fisher Scientific) and dispase (0.3 mg/mL; Gibco, Thermo Fisher Scientific) to DMEM GlutaMax (Thermo Fisher Scientific).

Prior to digestion, the pancreatic samples were placed in a Petri dish filled with DMEM GlutaMax, supplemented with 0.1% (*w*/*v*) bovine serum albumin (BSA, Gibco, Thermo Fisher Scientific) and 1% (*v*/*v*) pen-strep, hereafter referred to as DMEM 1%. The samples were then minced into small pieces using a sterile scalpel and transferred to a new 50 mL tube to remove the attached fat via repeated pipetting. The medium was discarded, and fresh DMEM 1% was added, and the cleaning steps were repeated until the suspension was clear. Subsequently, the fragments were resuspended in the collagenase–dispase enzyme media, and the tubes were gently mixed via inversion before being incubated at 37 °C in a shaking water bath until all the tissue fragments were digested.

The tubes were spun at 450× *g* for 5 min. All pellets were resuspended in DMEM 1% and were combined in one tube (individual dogs remained separated). The tubes were centrifuged again at 450× *g* for 5 min; the supernatant was removed; and the pellet was resuspended in the appropriate amount of Matrigel^TM^ (Corning, New York, NY, USA) with a concentration of, lot-dependent, 6–8 mg/mL and plated in in a 24-well plate or in a 12-well plate. The plate was placed in the incubator at 5% CO_2_ and 37 °C, allowing the Matrigel to solidify (Figure 1A). After 15 min, dog pancreas expansion medium (dpEM, see Section 2.1.3) was added. For the first 3 days the dpEM was supplemented with Rho kinase inhibitor Y-27632 (10 µM; Rock inhibitor, Stemcell Technologies, Vancouver, BC, Canada).

#### 2.1.3. Media Compositions, Expansion, Differentiation, and Maturation

The dpEM consisted of A/D+, 30% (*v*/*v*) of Wnt-conditioned media (Wnt3a), 10% (*v*/*v*) R-spondin-3 Fusion protein-conditioned medium (Immunoprecise Antibodies Ltd., Victoria, BC, Canada), 1% (*v*/*v*) B27 supplement without vitamin A (Invitrogen, Carlsbad, CA, USA), 1% (*v*/*v*) N2 supplement (Invitrogen), 10 mM nicotinamide (NIC, Sigma-Aldrich, Merck, St Louis, MO, USA), 1.25 mM N-acetylcysteine (NAC, Sigma-Aldrich, Merck), 100 ng/mL FGF10 (Peprotech, Rocky Hill, NJ, USA), 10 nM gastrin (Sigma-Aldrich, Merck), 100 ng/mL noggin (Peprotech), 50 ng/mL EGF (Invitrogen), 50 µg/mL primocin (InvivoGen, San Diego, CA, USA), and 0.5 µM A83-01 (Tocris Bioscience, Bristol, UK) [15,16]. The medium was changed two or three times a week, and passaging occurred every 7–10 days at a split-ratio between 1:6 and 1:8. All cultures were kept in a humidified atmosphere of 95% air and 5% CO_2_ at 37 °C (Figure 1B).

The differentiation was performed in multiple steps, mimicking the pancreatogenesis. This novel step-by-step differentiation protocol was based on previous publications leading towards (mature) β-cell differentiation in other mammalian species [17,18,19,20,21,22]. The differentiation of the organoids was divided into PDM-A (4 days of differentiation), PDM-B (3 days of differentiation), PDM-C (7 days of differentiation), PDM-D1 (7 days of differentiation), and lastly PDM-D2 (4 days of final maturation) (Figure 1C). For a detailed overview of all the media, with corresponding components and concentrations, see Appendix A.

#### 2.1.4. Islet of Langerhans Isolation

The protocol used for the islet isolation was based on Villareal et al., 2019, with some alterations [23]. Details on the animals used are in Appendix A. Briefly, after the digestion step and after the addition of the STOP solution, the tubes were spun down at 300× *g* for 1 min. For the washing steps, Hank’s Balanced Salt Solution (HBSS) was used for the first wash, and then it was replaced with phosphate-buffered saline (PBS) during the whole isolation.

### 2.2. RNA Isolation and qPCR

Total RNA was isolated with Qiagen columns and included an on-column DNase-1 treatment to minimize contamination with genomic DNA. cDNA was made using the iScript^TM^ cDNA synthesis kit, according to the manufacturer’s instructions (Bio-rad, Veenendaal, the Netherlands), with a mix of random hexamers and oligo-dT primers to guarantee optimal cDNA synthesis. A SYBR-Green (Bio-rad)-based RT-pPCR was performed in a 384-well format qPCR machine (CFX384 Real-Time System, Bio-Rad) [24]. To normalize the relative mRNA levels the reference genes, 40S Ribosomal Protein S5 (RPS5), Signal Recognition Particle Receptor (SRPR), and Succinate Dehydrogenase Complex Flavoprotein Subunit A (SDHA) were used. The details of the primer sequences, optimal annealing temperature, and amplicon sizes are listed in Appendix A. MIQE-precise guidelines were implemented as required [25].

### 2.3. Glucose-Stimulated Insulin Secretion Assay

The organoids were collected and made into single cells using TrypLE and washed twice with PBS. Then, the cells were acclimatized in Krebs–Ringer Solution, HEPES-buffered 1 mM glucose (Thermo Fisher, scientific) with 0.1% (*w*/*v*) BSA and Glucagon-Like Peptide 1 (GLP-1, Sigma-Aldrich, Merck) at a final concentration of 100 nM for 1 h. Subsequently, the cells were washed twice with PBS and transferred to a glucose-supplemented Krebs–Ringer Solution (final concentration 25 mM) for 1 h. After incubation, the cells were spun down at 125× *g* for 1 min. The supernatant was carefully collected, and insulin levels were measured with the Dog Insulin ELISA kit (Crystal Chem, IL, USA), according to the manufacturer’s instructions. A multimode plate reader (excitation/emission at 450/630 nm) was used, and the results were used to normalize the cell count.

### 2.4. Statistical Analysis

Data analysis was executed in Graphpad using the Student’s *t*-test on data sets with independent groups for qPCR and the paired Student’s *t*-test on the data set for the GSIS. *P*-values below or equal to 0.05 were considered significant. *P*-values were indicated as *, **, and ***, respectively, meaning *p* ≤ 0.05, *p* ≤ 0.01, and *p* ≤ 0.001. The mean of at least triplicates was used.

## 3. Results

### 3.1. Expansion, Differentiation, and Maturation of dPO

The dog pancreas organoids (dPOs) were cultured in Matrigel droplets and maintained in expansion media (dpEM Figure 2A) before being differentiated towards insulin-producing β-cells. Differentiation was initiated using a series of pancreas differentiation media (PDM), including PDM-A, PDM-B, PDM-C, and PDM-D1, designed to mimic pancreas organogenesis. Subsequently maturation was performed with PDM-D2 (Figure 2B–F). Upon exposure to PDM-D1, the cells began to dissociate from the organoid and clustered together, forming islet-like structures with a characteristic golden hue (Figure 2G,H).

### 3.2. Gene Expression of Stem Cell Markers, β-Cell Markers, and Non-β-Cell Markers

Relative gene expression measurements were performed in order to molecularly describe the step-wise differentiation process. To this end, the relative expression of specific markers, such as *PDX1* and *NKX6.1*, which are expressed early during organogenesis, was measured in expansion medium (dpEM), differentiation medium B (PDM-B), and differentiation medium D (PDM-D2). To relate these data to a clinical perspective, isolated canine islets were included as a reference. A small increase in PDM-B was observed, and in PDM-D2, the relative expression levels were similar to those observed in isolated islets (Figure 3A,B). This observation suggested a strong enrichment of β-cells under PDM-D2 culture conditions. To explore this further, we analyzed relative gene expression of makers of other cell types in the pancreas, such as *LGR5* and *SOX9* (both stem cell markers); *GCG* (alpha-cell marker); *SST* (delta-cell marker); and *GLUT2, INS,* and *PCSK1* (glucose transporter, insulin, and proprotein convertase 1, typical β-cell markers) (Figure 4A–G). Two classical stem cell markers, *LGR5* and *SOX9*, were clearly detectable under EM culture conditions but showed a dramatically lowered expression in DM culture condition (around 20-fold lower expression). The low expression of *GCG* (*glucagon)* and *SST* (*somatostatin)* indicated a lack of alpha- and delta-cells in the organoids cultured in the final PDM-D2 medium (Figure 4F,G). Although the relative expression of *INS* (*insulin*) increased strongly in PDM-D2 compared to expansion medium, the levels were still substantially lower than those observed in isolated islets (Figure 4C). In contrast, the relative expression of *GLUT2* and *PCSK1* increased in PDM-D2, and for *PCSK1*, the relative expression increased even to islet levels (Figure 4A,B). Relative *INS* levels remain around 30% of the islet expression.

To standardize the differentiation protocol as much as possible, we used female Beagle dogs for this purpose only. From a theoretical point of view, a comparison with islets for female Beagle dogs would have been preferred, but it would likely not commence in a completely different conclusion on the improved differentiation in differentiation medium. A.

### 3.3. Glucose-Stimulated Insulin Secretion Assay

Given that most of the markers indicated an enrichment of β-cells under the final differentiation culture conditions, and considering that insulin release is mainly regulated at the (proteolytic) post-transcriptional level, we measured insulin release into the medium following glucose stimulation of glucose starved cells (Figure 5). Cells starved for as little as 1 h under low-glucose conditions release around 0.9 ng/mL per 100,000 cells, while glucose stimulation resulted in insulin release of 1.5 ng/mL per 100,000 cells.

## 4. Discussion

This study shows that canine pancreatic ductal cells can serve as a cell source for differentiated β-cells when sequential culture media designed to mimic pancreas organogenesis are used. This differentiation protocol successfully resulted in glucose-responsive β-cells. To standardize the differentiation protocol as much as possible, we used female Beagle dogs for this purpose only. From a theoretical point of view, a comparison with islets for female Beagle dogs would have been preferred, but it would likely not commence in a completely different conclusion on the improved differentiation in differentiation medium. As such, these organoids can be used for disease modelling studies aimed at long-term improvement in the quality of life for dogs suffering from DM1.

Several factors may account for the good differentiation and maturation observed, but only leading to moderate glucose-induced insulin release. First, inherent to working with primary cells is the large variability between donors, as indicated by the variations in gene expression levels and insulin release. Second, most recombinant growth factors were of human origin. While the fundamental processes of organogenesis are highly similarly regulated across mammalian species, it is possible that full maturation was not achieved in canine-derived cells due to species-specific differences in protein activity. Last, it is possible that the 1 h starvation period under low-glucose conditions may not have been sufficient to completely abolish insulin release before glucose stimulation.

The differentiation media are designed based on reports on pancreatic embryogenesis, since little information is known about the self-regeneration capacity of the pancreas. It is uncertain whether similar pathways are also effective in the differentiation of undifferentiated adult pancreatic ductal cells. PI3K/Akt/mTOR and JAK2/STAT5 pathways are known to be involved in regulating the regeneration of the endocrine pancreas. Furthermore, hepatokines, adipokines, gut hormones, and lactogens significantly impact β-cell regeneration and β-cell mass [26]. While these pathways have not been investigated in the context of pancreatic organoids, it is possible that their targeted modulation could lead to improved functionality.

One recent important study used the same cell source (ductal cell) and animal model (canine) as this study [27]. However, instead of using cadaver pancreases of adult dogs, they used pancreases from five newborn pups. These pups were selected because of dystocia during parturitions; since this occurs rarely (5% in dogs [28]), it is not the most reliable method to acquire material. Moreover, depending on the passage number, these organoids consisted of around 80% insulin positive and 15% glucagon positive cells. This contrasts with our cells, which seem to be highly enriched in β-cells, without any *glucagon* gene expression. The levels of insulin release also varied significantly. In the other study, organoids released in high-glucose 0.08 ng/mL/10^5^ cells under high-glucose conditions and 0.015 ng/mL/10^5^ cells under low-glucose conditions. In contrast, the organoids in our study exhibited much higher insulin release levels, viz. 2.02 ng/mL/10^5^ cells in high-glucose conditions and 0.65 ng/mL/10^5^ cells under low-glucose conditions. These differences are biologically significant and suggest improved functionality in our organoids. Interestingly the relative INS mRNA levels (Figure 4) were below the islet levels. Release from pre-existing vesicles loaded with insulin protein can explain the difference, but the lower relative mRNA levels provide room for further improvement.

To mimic in vivo conditions more closely, Glucagon-Like Peptide 1 (GLP-1) was added. GLP-1 is known to increase intracellular cyclic AMP (cAMP) levels and inositol trisphosphate (IP_3_), both resulting in the release of calcium from the endoplasmic reticulum (ER). Both are able to enhance protein kinase A (PKA) and PKC activity to aid in granule trafficking in first- and second-phase insulin secretion, respectively [29]. In contrast to glucose, GLP-1 alone cannot trigger insulin secretion. However, during food intake, it enhances glucose-induced insulin secretion, allowing the β-cells to optimally adapt to the metabolic situation, thus better mimicking an endogenous environment. Whether this indicates that improved β-cell functionality can be achieved by co-culture with glucagon-producing α-cells remains an open question but sounds reasonable.

### Conclusion and Future Perspectives

The step-by-step differentiation protocol resulted in β-cells secreting insulin levels suitable for disease modelling. However, this in vitro culture system is highly artificial, and clinical application, as cell replacement therapy still has a long way to go for numerous reasons. Despite a recent successful treatment of a patient with iPSC-derived islets, challenges remain ([30], and recently reviewed ([31]). First, this study made use of cells obtained from healthy animals. It remains to be seen if a similar differentiation protocol on ductal cells harvested from DM1 dogs will also result in equal amounts and similarly differentiated β-cells. Second, the culprit of DM1, immune-attack on β-cells, remains intact. Therefore, additional safeguards are needed to avoid immune-mediated degradation of the transplanted β-cells-based organoids. Lastly, this differentiation protocol made use of cells cultured on a Matrigel-based scaffold. This material is animal-derived and, as such, unsuitable for clinical applications. Moreover, the fact that Matrigel is derived from inoculated tumors in mice raises serious ethical concerns regarding the utilization of Matrigel. All of these hurdles hamper applications in cell replacement strategies, even irrespective of the enormous costs associated with cell-replacement therapies. Recently, a combination with canine Mesenchymal Stromal Cells (MSCs), having immunomodulatory features, to create neo-islets provided a significant step towards clinical cell replacement application of these differentiated β-cells [32]. However, the combination of our organoids with improved β-cell differentiation and MSCs has not yet been explored. Therefore, the most immediate and viable application of the organoids with improved β-cell differentiation lies in the field of disease modelling.

Whereas the application of these organoids for transplantation seems too farfetched as of now, for species-specific disease modelling, dissecting the mechanisms of fatty pancreas, a recently described phenomenon, is a short-term realistic target [33,34]. Although fatty pancreas has not been described in veterinary medicine, given the similarities between human and canine DM1, it can be anticipated that the disorder also presents in dogs. Our experience in lipid loading of liver organoids in the past [35,36] makes this disease an interesting one to model in organoids.

## Figures and Tables

**Figure 1 vetsci-12-00362-f001:**
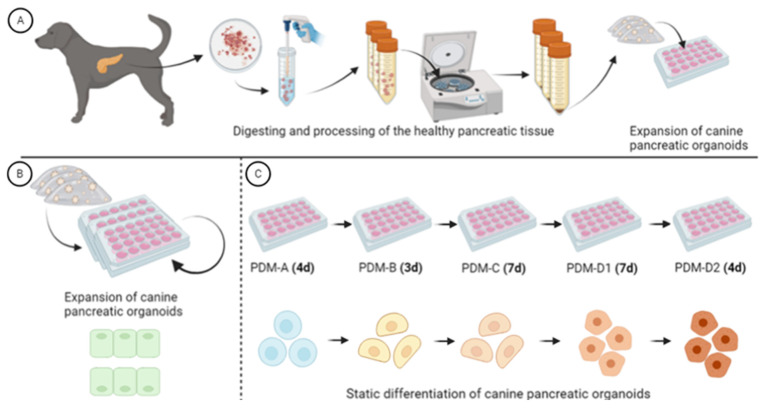
Experimental design. (**A**) Schematic overview of the isolation, digestion, and processing of pancreatic tissue samples. (**B**) Expansion of the canine pancreatic ductal organoids before the differentiation. (**C**) Differentiation of the canine pancreatic ductal organoids into immature β-cells with the corresponding differentiation media. Pancreas differentiation media (PDM). Created with Biorender.com.

**Figure 2 vetsci-12-00362-f002:**
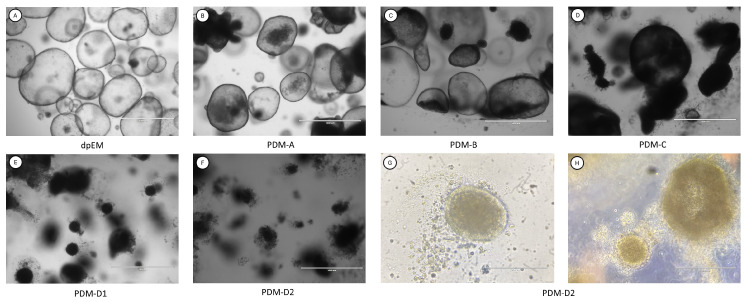
Morphology of dog pancreas organoids (dPOs) on different time points prior to media-switch media change. Brightfield pictures of the morphology of dPOs in EM and PDM; scale bar = 1000 µm (**A**–**F**). In addition, brightfield pictures in color of the morphology of dPOs in PDM-D2; scale bar = 200 µm in first picture (**G**), and scale bar = 1000 µm in second picture (**H**).

**Figure 3 vetsci-12-00362-f003:**
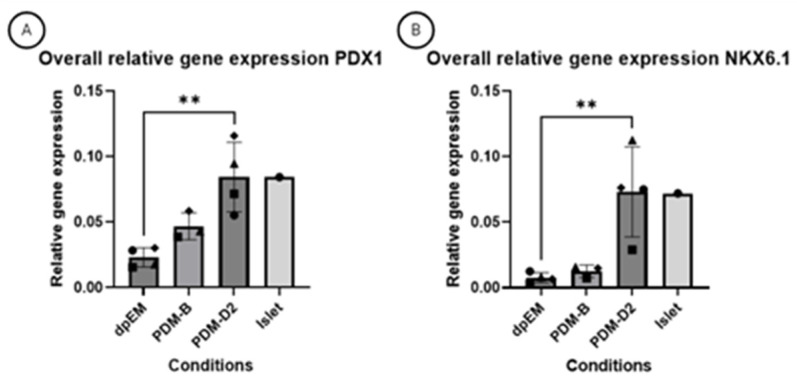
Rt-qPCR of specific markers PDX1 (**A**) and NKX6.1 (**B**) from four independent donors in dog pancreas expansion media (dpEM), pancreas differentiation media B (PDM-B), pancreas differentiation media D2 (PDM-D2), and one sample of an isolated islet of an independent donor. The plotted bars represent the mean of the group, with error bars representing the standard deviation. Normalization of the mRNA was performed with the reference genes 40S Ribosomal Protein S5 (RPS5), Signal Recognition Particle Receptor (SRPR), and Succinate Dehydrogenase Complex Flavoprotein Subunit A (SDHA). Boxes, triangles, circles, and diamonds represent individual donors. Two asterisks indicate significance *p* ≤ 0.01.

**Figure 4 vetsci-12-00362-f004:**
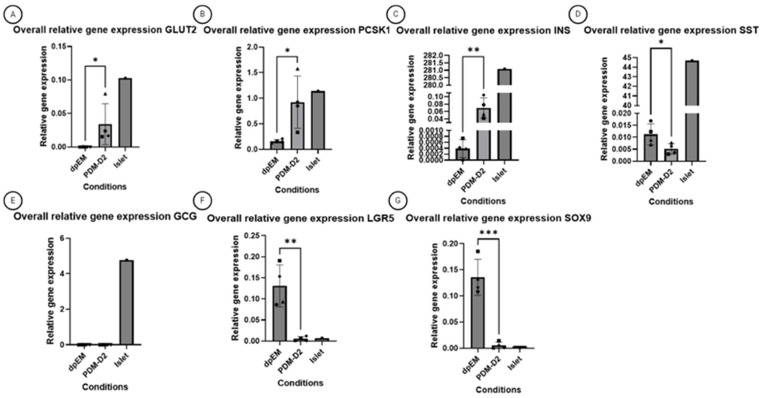
Rt-qPCR of several genes of interest from four independent donors in expansion media and differentiation media and one sample of an isolated islet of an independent donor. The plotted bars represent the mean of the group, with error bars representing the standard deviation. (**A**) GLUT2; (**B**) PCSK1; (**C**) INS; (**D**) SST; (**E**) GCG; (**F**) LGR5; (**G**) SOX9 Normalization of the mRNA was performed with the reference genes 40S Ribosomal Protein S5 (RPS5), Signal Recognition Particle Receptor (SRPR), and Succinate Dehydrogenase Complex Flavoprotein Subunit A (SDHA). Boxes, triangles, circles, and diamonds represent individual donors. * Indicates significance *p* ≤ 0.05; ** and *** mean *p* ≤ 0.01, and *p* ≤ 0.001, respectively.

**Figure 5 vetsci-12-00362-f005:**
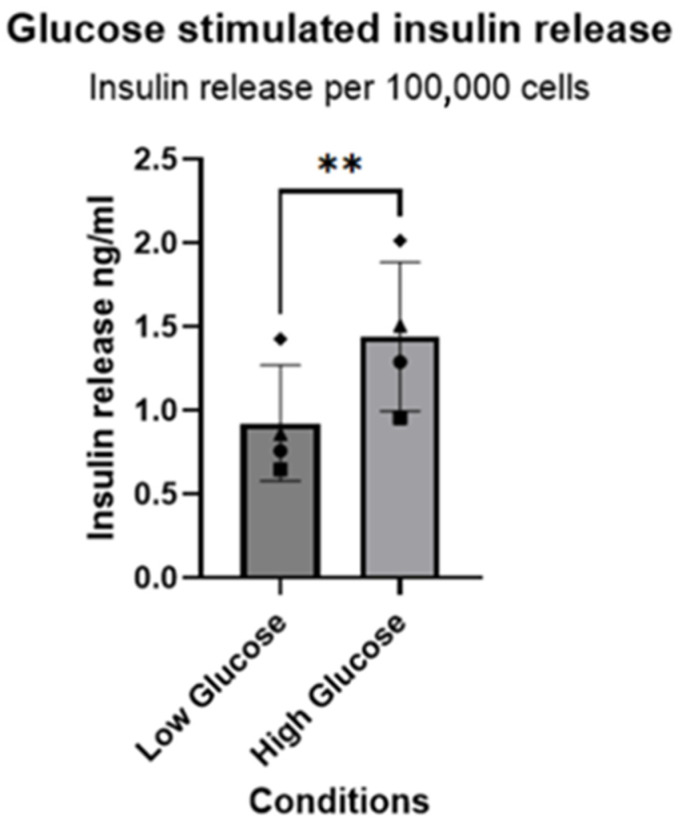
Glucose-stimulated insulin secretion (GSIS) assay of the four independent donors used in differentiation media. Glucose concentration started with 1 mM in low-glucose solutions and ended with high-glucose concentrations of 25 mM. In addition, Glucagon-Like Peptide 1 (GLP-1) was added to mimic a more endogenous process. To normalize the results, the cells were counted and displayed as insulin release per 100,000 cells. Diamond, pyramids, circles and squares represent different donor. ** indicate significance *p* ≤ 0.01.

## Data Availability

No new data were created or analyzed in this study. Data sharing is not applicable to this article.

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
