# Peer review of "Improved Differentiation Towards Insulin Producing Beta-Cells Derived from Healthy Canine Pancreatic Ductal Organoids"

_vetsci, 2025, doi:10.3390/vetsci12040362_

Round 1

Reviewer 1 Report

Comments and Suggestions for Authors

The paper is well written and the results are clearly presented, the differentiation protocol that was employed seemed to work however insulin levels were low and I don't think would be useful for curing diabetes at this stage. This is recognised by the authors. I gather the authors are only wishing to use these cells in dogs and not humans as the pig islet xenotransplantation group have been working towards for decades. The discussion of the paper could better develop the possible usefulness of these cells as tools  for experimentation and this would make the results more interesting to the general audience.

Author Response

detailed reactions are ion pdf below

Reviewer 2 Report

Comments and Suggestions for Authors

The study by Gouw et al established a protocol to differentiate pancreatic ductal cells from healthy dogs towards insulin-releasing beta cells. The purpose is clinically relevant, given the lack of valuable current options for diabetes treatment in dogs.

Comments:

  1. One major issue is the fact that the authors used healthy donors. Despite this is mentioned as a limitation in Conclusion, it would be important to further stress it. Type 1 diabetic donors would be important for autologous use. But it would probably provide a lower number of beta cells. This point must be further addressed, maybe in the title (“…derived from healthy canine pancreatic cells”), and/or Abstract and Introduction.
  2. The purpose of the second paragraph (line 32) is confusing. “Dogs are men’s best friend”, therefore present similar pathologies and need to be treated? Dogs are used for beta cell research purposes for long time given their similarity to pancreatic human physiology? The whole paragraph does not add relevance to the aim of the study and should be removed.
  3. Pancreatic islets were used for gene expression levels control. Islets were isolated from a different breed (bastard) from the experimental samples (Beagles). As the authors report, since the number of pancreatic cells can be influenced by breed, age and gender, it is important to tackle why female Beagle were not used to isolate pancreatic islets. Can this be a limitation of the study? Would you expect islets from a male bastard dog had higher/lower gene expression?
  4. Findings regarding GLUT2 and PCSK-1 gene expression were not described in Results. They can further be linked to results presented in Fig 5.
  5. Fig 4C shows that insulin levels increase with differentiation medium but are still substantially lower than in islets. These findings must be clearly explained. Although the authors comment the putative need of other types of cells to increase insulin release in Discussion (line 280), it is not clear whether they attribute the lower insulin released levels to the absence of these cells.
  6. In line 202, please correct Figure 43-3B.

Author Response

see pdf below

Reviewer 3 Report

Comments and Suggestions for Authors

The article by Gouw et al. provides an extensive description of differentiation of canine ductal organoids towards insulin producing beta cells. The experiments are overall well designed, and the conclusions agree with previous reports. Although this may hinder the originality of the paper the data reported are relevant since this seems still to be a viable approach in the field of disease modelling..

There are some aspects of the manuscript that require attention.

  • The paragraph from line 33 to 38 is not relevant for the content of the paper.
  • Line 48: “in men” should be “in humans”.
  • Lines 55 -57. The main therapy for DM in veterinary is Insulin. Although the paper provides compelling data regarding Regenerative Medicine it is opinion of this Reviewer that it won’t be any soon a viable alternative to insulin.
  • It is not clear why GLP1 was used for GSIS.
  • Why did the authors perform GSIS with dissociated organoids?
  • 5: It would be useful to provide a comparison with canine islets.

Overall, the paper is well written, and the data provided are solid. I particularly appreciated the honesty of the authors regarding the limitations of this study. This, however, doesn’t hinder the scientific merit of it.

Author Response

de tailed point-by-point in pdf
